# Effect of the Pellet and Mash Feed Forms on the Productive Performance, Egg Quality, Nutrient Metabolism, and Intestinal Morphology of Two Laying Hen Breeds

**DOI:** 10.3390/ani11030701

**Published:** 2021-03-05

**Authors:** Yi Wan, Ruiyu Ma, Anam Khalid, Lilong Chai, Renrong Qi, Wei Liu, Junying Li, Yan Li, Kai Zhan

**Affiliations:** 1Anhui Key Laboratory of Livestock and Poultry Product Safety Engineering, Institute of Animal Husbandry and Veterinary Medicine, Anhui Academy of Agriculture Science, No. 40 Nongke West Road, Hefei 230031, China; eternalwan@163.com (Y.W.); 18655160470@163.com (R.M.); easona123@126.com (R.Q.); liuweicau@163.com (W.L.); brandon2007@163.com (J.L.); Liyan1314526@126.com (Y.L.); 2College of Animal Science and Technology, Anhui Agricultural University, Hefei 230001, China; anamkhalid166@hotmail.com; 3Department of Poultry Science, University of Georgia, Athens, GA 30602, USA; lchai@uga.edu

**Keywords:** laying hen, feed form, production performance, nutrient metabolism, intestinal morphology

## Abstract

**Simple Summary:**

Although there is an established understanding of the nutritional requirements for poultry production, the evaluation of the feed form for chickens still needs to be further examined. It is generally believed that a pellet diet has a positive effect on chicken growth, but there are some studies that have reported no difference between pellet and mash on chicken’s performance. The present study indicated that pellet diets improved the production performance and nutrition metabolism through positive changes in the laying rate, feed intake, egg albumen quality and apparent digestibility of laying hens.

**Abstract:**

One of the most important factors that determine feed utilization by chickens is the feed form. Although it is generally believed that pellet diets have a positive effect on chicken growth, there are some studies that have indicated no difference between pellet and mash on chickens performance. This study was conducted to assess the effects of feed form on production performance, egg quality, nutrient metabolism and intestinal morphology in two breed laying hens. Two hundred and sixteen 25-week-old Hy-Line brown (*n* = 108) and Hy-Line grey (*n* = 108) hens were selected. Each breed was randomly allocated into two treatments with 6 replications (9 birds in each replication), which were fed mash and pellet diets, respectively. Production performances were recorded daily and egg quality traits were measured every two weeks. At 42 weeks of age, one bird per replication from each experimental group was selected for metabolism determination and intestine morphology observation. Compared with mash diets, pellet diets improved laying rate (*p* < 0.05), ADFI (average daily feed intake, *p* < 0.05), egg weight, shell strength, yolk proportion and Haugh unit (*p* < 0.05) in both breeds and reduced the FCR (feed conversion ratio, *p* < 0.05) in Hy-Line grey. The apparent digestibility of DM% (dry matter) and CP% (crude protein) were significantly higher (*p* < 0.05) in both breed laying hens fed pellet than those fed mash. The apparent digestibility of P% (phosphorus) and Ca% (calcium) was higher in Hy-Line grey fed pellet and was higher in Hy-Line brown fed mash. Compared to mash diets, pellet diets increased the VH (villus height), CD (crypt depth) and VCR (ratio of villus height to crypt depth) of the small intestine of Hy-Line grey, and increased the VH and CD of duodenum and ileum of Hy-Line brown. Overall, pellet diets improved production performance and nutrition metabolism through positive changes in the laying rate, feed intake, egg albumen quality and apparent digestibility of laying hens. The current findings provided support for the advantages of feeding pellets during the peak egg laying period for the two popular laying hen strains, Hy-Line brown and Hy-Line grey.

## 1. Introduction

Laying hen nutritionists and the egg industry have been studying new ways to optimize the utilization of feed and production efficiency [1,2]. The feed form plays a critical role in determining growth performance, digestion, nutrient digestion, intestinal health, and productive performance of poultry [3,4,5]. Mash and pellet are two commonly used forms of chicken feeds in commercial egg production. Mash (smaller particle size) is a complete feed form that is finely ground and mixed so that ingredients cannot be separated easily [6]. Pelleting is a step in feed processing in which the ingredients are agglomerated by mechanical action, in combination with moisture, pressure and temperature, to form larger structures called pellets [2].

Pellet diets were reported to have positive effects on growth performance, feed intake, and feed conversion ratio of chickens [7,8]. Other benefits of pelleting include the reduction in ingredient segregation, ease of handling, improved feed flow in the equipment, and reduction in formulation cost by including alternative ingredients and decreasing the diet energy [2]. In contrast, some reports suggested that pellet diets may even decrease nutrient utilization and starch digestibility of broilers under some conditions [9]. On the other hand, mash diets were reported to improve the feed conversion ratio, enhance starch digestibility and improve intestinal glucose uptake of broilers compared to those fed pellet diets [10]. However, in a few recent studies, it was reported that laying hens from white and brown strains did not respond significantly to manipulations in pellet or mash diets in terms of productive performance [11,12]. 

In the present study, we hypothesized that the feed form may be beneficial for nutrient utilization in laying hens and subsequently affect laying performance. Therefore, pellet and mash feed forms were used in standard commercial diets to investigate their effects on production performance, egg quality, nutrient metabolism and intestinal morphology of two breed laying hens during the peak laying period (i.e., 26 to 43 weeks of age).

## 2. Materials and Methods

The study protocol was approved by the Committee for the Care and Use of Experimental Animals at Anhui Academy of Agricultural Science under permit No. A11-CS06.

### 2.1. Animals and Management

Two hundred and sixteen 25-week-old healthy Hy-Line brown (*n* = 108) and Hy-Line grey (*n* = 108) laying hens with similar body weights were selected. All birds were raised in individual laying cages subjected to a light/dark cycle (16 h light: 8 h dark) and given free access to feed and water. Feed was supplied in troughs placed in front of each cage twice a day, and water was provided from nipple drinkers continuously. Feed was not completely consumed by birds between feedings, and the total amount of feed for each time was according to the birds’ daily demand to ensure adequate supply. The average indoor air temperature and relative humidity were controlled at 20.3 °C and 56%, respectively, during the trial period.

### 2.2. Diets and Feeding

The basal diets (Table 1) were formulated to meet the requirements for layers recommended by the Agricultural Trade Standardization of China (NY/T33–2004), and were made into mash compete feed. Mash compete feed with 5% water added was well mixed by the leaf belt horizontal spiral mixer (Wuxi Taihu Geain Machinery Co., Ltd., Wuxi, China) for 5 min, and pelleted by a particles granulator (9KS-280, Xinhua feed factory, Yuyao, China) at 65 °C. Each time the diets were made, they were mixed as one basal diet that was then simply split, with one half remaining in mash form and the other half being pelleted. Pellet and mash diets had similar feed ingredients and nutrient contents. Considering the added water would increase the moisture content of the pelleted diet, feed consumption data were adjusted for the increased water weight of the pelleted diet.

The physical parameters of mash and pellet are shown in Table 2 and Table 3, respectively. The bulk density was measured according to Aarseth et al. [13]. The repose angle and size distribution of mash were measured according to Ileleji and Zhou [14]. The moisture content of pellet was measured according to Lam and Flores [15]. Hardness was measured in a hardness tester (Nova Etica, model 298 DGP-Ethiktechnology, Sao Paulo, Brazil) using individual pellets (20 pellets per treatment). The water resistant time was determined according to Obaldo et al. [16]. To determine the finely ground (powder-like) content of the pellet, three replicates (200 g of each) were weighed, sieved through a 4 mm sieve for 30 s, and the feed remaining in the sieve was again weighed to calculate the percentages of intact pellets and fines.

Each layer breed (i.e., 108 Hy-Line grey and 108 Hy-Line brown) was randomly allocated into two groups (54 hens each group), which were fed mash and pellet diets, respectively. There were six replications for each group (9 hens per replication). There was a one-week preliminary feeding period for the mash group (as hens were fed pellet diets prior to the acclimation period), and then the experimental period lasted for 18 weeks (hens were 26 to 43 weeks of age).

### 2.3. Production Performance

Eggs were collected, counted, and weighed every day to calculate the laying rate, average egg weight and the ratio of abnormal eggs. ADFI (average daily feed intake) and the FCR (feed conversion ratio) were recorded/calculated for analyzing production performance. FCR was calculated as the ratio of feed intake per unit of egg mass.

### 2.4. Measurement of Egg Quality

From 26 weeks of age, 30 eggs were randomly sampled from each group for egg quality analysis every two weeks. All eggs were kept in the same storage room and egg quality measurements were completed on the day of collection. Measurements of egg length and width were taken with a digital caliper to the nearest 0.01 mm, and egg shape index was the ratio of length to width. Egg weight and shell weight were measured using an electronic scale with an accuracy of 0.01 g. Shell strength was measured with an eggshell force gauge (EGG-0503, Robotmation Co., Ltd., Tokyo, Japan). Shell thickness was measured at the eggshell equator in three places using a micrometer gauge (FHK Co., Ltd., Tokyo, Japan). Yolk color, albumen height and Haugh unit (HU) were measured using an automatic egg multitester (EMT-5200, Robotmation Co., Ltd., Tokyo, Japan). Yolks were separated from the albumen, and then the chalazae were carefully removed from the yolk before weighing the yolk. The percentages of yolk and shell were quantified based on the following equations: Egg shell (%) = 100 × (shell weight/egg weight)(1)
Egg yolk (%) = 100 × (yolk weight/egg weight)(2)

### 2.5. Nutrient Utilization and Determination

At 42 weeks of age, one bird per replication from each experimental group was randomly selected for metabolism determination by the total excreta collection method. There was a three-day preliminary trial period to record the average daily feed intake and to get familiar with the regularity of animal defecation. There was a 24 h period for birds to empty the original excreta in the intestines prior to the formal experiment. In the formal experimental period (2 days), each bird was fed with 80% of recorded feed intake twice a day. Total excreta of each bird were collected and weighed once a day. Every 100 g fresh excreta were added to 10 mL 10% hydrochloric acid and stored in an ice box to prevent loss of ammonia nitrogen. The moisture of excreta was regained under room temperature for 24 h after drying at 65 °C in the drying oven for 8 h. The dry samples were grond and sifted out (40 mesh), and then were kept in a self-sealing bag under cold preservation. The content of dry matter (DM), crude protein (CP), calcium (Ca) and total phosphorus (P) was determined using the oven-drying method, Kjeldahl method, ethylene diamine tetraacetic acid (EDTA) complexometric titration and vanadium molybdate yellow colorimetric method, respectively. The apparent digestibility was calculated according to the following equation: Apparent digestibility (%) = (nutritive substance from ingestion (g) − nutritive substance in the excreta (g))/nutritive substance in the feed (g) × 100(3)

### 2.6. Morphology of Small Intestine

At 42 weeks of age, one bird per replication of each experimental group was randomly selected for intestine morphology observation. Segments of about 3 cm in length of the middle portion from the duodenum, jejunum and ileum (from the vitelline Meckel’s diverticulum to 4 cm above the ileo–cecal junction) were removed from the small intestine. Removed segments were flushed with a 10% neutral buffered formalin solution and were then used for morphometric analysis. For morphometric analysis, segments were fixed in a 10% neutral buffered formalin solution for 24 h. Intestinal samples were then excised, dehydrated in a tissue processing machine (Leica Microsystem K. K., Tokyo, Japan) and embedded in paraffin wax. Sections of 4 mm were cut from each sample, fixed onto slides, stained with haematoxylin and eosin, mounted and examined under the light microscope. Stained slides were observed under a Motic BA210, visual measurement of villus height (VH) and crypt depth (CD) were performed at 10× (objective lens) with image software (Motic Image Plus 2.0^ML^ Soft, Motic China Group Co., Ltd., Xiamen, China). VH was measured from the crypt–villus junction to the brush border at the tip. CD was taken at the level of the basement membranes of opposing crypt epithelial cells. The ratio of villus height to crypt depth (VCR) was calculated.

### 2.7. Statistical Analysis

Data were analyzed by two-way ANOVA to determine the main effects (feed form and layer breed) and their interaction using the General Linear Model (GLM) command in SAS version 9.3 statistical software (SAS Institute Inc., Cary, NC, USA). Tukey’s multiple comparison was used to test the significance of the differences between treatment means; significance was declared at *p* < 0.05. All data are presented as the mean and standard error of the mean.

## 3. Results

### 3.1. Production Performance

Effects of feed form on the production performance are shown in Figure 1 and Table 4. Birds fed pellet diets had a higher laying rate and ADFI in both breed laying hens than those fed mash diets, but had a significantly lower FCR (*p* < 0.05) in Hy-Line grey hens, which showed positive effects on the feed conversion. However, the FCR was lower in Hy-Line brown hens fed mash diets than those fed pellet diets. The ratio of abnormal eggs was slightly decreased in both breeds fed pellet diets than mash diets. No influence of feed form was observed for the mortality in both breeds. There was an interaction (*p* < 0.01) between feed form and breed for laying rate and the FCR.

### 3.2. Egg Quality Traits

Effects of feed form on the egg quality traits are shown in Table 5. Compared to mash diets, pellet diets increased the egg weight (*p* < 0.05), shell strength, yolk proportion and Haugh unit in both breeds, but slightly reduced shell proportion, yolk color and shell thickness of Hy-Line grey hens. No obvious difference between the two feed form groups was observed in both breeds for the egg shape index. There was no interaction (*p* > 0.01) between feed form and breed for the egg quality traits except for the shell strength and yolk color.

### 3.3. Apparent Digestibility

Effects of feed form on the apparent digestibility of birds’ nutrients are shown in the Table 6. The apparent digestibility of DM% and CP% increased by 2.73% and 4.77%, respectively, in Hy-Line grey hens fed with pellet diets (*p* < 0.05), and increased by 2.18% and 4.30%, respectively, in Hy-Line brown hens fed with pellet diets (*p* < 0.05). The level of P% and Ca% was higher in Hy-Line grey hens fed with pellet diets and in Hy-Line brown hens fed with mash diets. There was no interaction (*p* > 0.01) between feed form and breed for the apparent nutrient digestibility.

### 3.4. Small Intestinal Morphology Structure

Effects of feed form on the morphological parameters of small intestine are shown in Table 7. Compared to mash diets, pellet diets significantly increased (*p* < 0.05) the VH and VCR of duodenum and ileum in Hy-Line grey hens. Similarly, higher VH and CD of duodenum and ileum, as well as a higher VCR of duodenum, were observed in Hy-Line brown hens that were fed pellets compare to those fed with mash diets. No significant difference was observed for the morphological structure in the jejunum (*p* > 0.05) between the two feed form groups in both breeds. There was an interaction (*p* < 0.01) between feed form and breed for the VH of the duodenum and ileum.

## 4. Discussion

Laying rate and feed conversion ratio are regarded as the main evaluation factors for layer performance. It has been well reported that pelleting of feeds enhanced the economics of production by improving the laying rate and feed efficiency responses in chickens [7]. However, a few studies [17,18] suggested that there was little difference between pellets and mash diets on chicken performance, especially during the late phase of production. In the current study, the egg laying rate of both Hy-Line grey and Hy-line brown hens fed with the pellet diet were higher than those fed the mash diet, which was consistent with Morgan and Heywang’s study [19]. Higher daily feed intake was found in both breeds fed with pellets, which is similar to previous reports [20,21] that concluded that chickens tend to consume more pellets than mash diets. The higher feed intake for pellet groups may be caused by the process of pelleting in which steam and mechanical pressure are applied to the mash to agglomerate the feed particles and improve the texture of the feed [22]. Consequently, pelleting diets increase bulk density and facilitate feed intake. The pelleting process has a higher cost compared to the mash, but this cost can be compensated by increased growth performance, especially in broilers [23]. Mortality was not affected by feed form according to the current study, which agrees with the findings in Hamilton and Proudfoot (1995) [24] that the mortality of laying hens was not influenced by the particle size or form of diets. Although there were interaction effects between feed form and breed for laying rate and feed intake, the results within each breed were consistent in supporting pellet diets.

The type of feed form did not have a significant effect on most egg quality traits, which agrees with the results reported by Ege et al. (2019) [25] that there was no significant effect on egg quality traits except for yolk color score by alterations in particle size or feed form. Zheng et al. (2020) [26] also reported that egg quality characteristics were little affected by feed form (mash vs. pellet). The egg weight and Haugh unit were higher in both breeds of hens fed with pellet diets compared to those of the mash diets in the current study, which showed a higher level of albumen quality. Higher egg weight and Haugh unit in the pellet group may be attributed to higher feed intake and protein metabolism of birds. 

Higher apparent metabolism of DM and CP were found in pellet groups in the present study, similar to the results in Hetland et al. (2002) [27] that show larger particle size of feed could stimulate gizzard activity and promote intestinal peristalsis, resulting in higher metabolism of nutrients compared to birds fed with fine feed particles. In contrast, it was hypothesized that fine grinding increased the available surface area for the activity of endogenous enzymes and thus improves the nutrient digestibility of birds [28]. In the evaluation of different energy levels in pelleted diets, Massuquetto (2018) [21] found a reduction in the digestibility of DM and CP for broilers fed with mash diets.

The small intestine is the main organ in the gastrointestinal tract [29], its structure supports the digestion and absorption of nutrients such as VH, CD and VCR, which can reflect the functional status of small intestine [30]. The longer VH might be combined with increased surface area and hence greater absorption [31]. Higher VCR indicated higher intestinal secretory ability and may lead to greater nutrient digestibility and growth performance of chickens [32]. In the current study, a greater VH and CD in the small intestine, as well as a higher VCR in the duodenum, were observed in both breeds of hens fed with pellets than mash. This histomorphological finding aligns with the results of Abadi et al. (2019) [5] that show birds fed with pelleted diets had a greater value of VH and VCR in the duodenum and jejunum in male Ross 308 chickens than those fed with mash diets. Dahlke et al. (2003) [33] also reported that reduction in the particle size of the diets led to shallower crypts and shorter duodenal villi. Higher VH and CD in the small intestine of birds fed pellet was thought to be caused by the higher feed intake and higher flow of nutrients in the proximal small intestine [28]. Interactions between feed form and breed for the VH of duodenum and ileum indicated that changes in the morphological structures could be breed-specific when feeding different feed forms.

## 5. Conclusions

The current study tested the effect of the pellet and mash feed forms on the productive performance, egg quality, nutrient metabolism, and intestinal morphology of two laying hen breeds. During the peak laying period (i.e., 26–42 weeks), pellet diets enhanced the egg laying rate, ADFI, egg weight, and improved egg albumen quality and the metabolism of DM and CP. The current findings provided support for the advantages of feeding pellets during the peak egg laying period for two popular laying hen breeds—Hy-Line brown and Hy-Line grey laying hens. Further studies are needed to evaluate the effects of pellet and mash feed form on performances and digestive tract parameters in larger populations with different chicken breeds.

## Figures and Tables

**Figure 1 animals-11-00701-f001:**
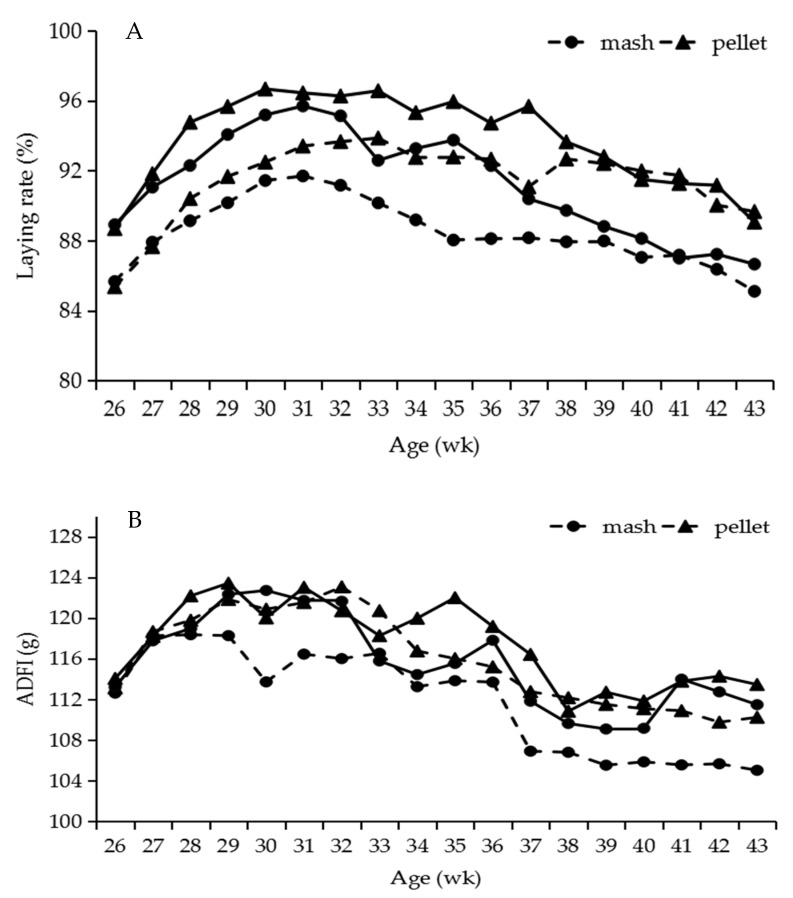
Laying rate (**A**) and ADFI (**B**) of two breed laying hens fed pellet and mash diets from 26 to 43 weeks of age. ADFI = average daily feed intake. Dotted lines represent Hy-line grey, solid lines represent Hy-line brown.

**Table 1 animals-11-00701-t001:** Ingredient composition and nutrient levels of the basal diets.

Ingredients, g/kg		Calculated Nutrient Level, %		
		Feed Form	Mash	Pellet
Corn	604.2	ME, MJ/kg	11.30	11.30
Soybean meal meal	255.8	Crude protein	16.66	16.69
Soybean oil	10.0	Calcium	3.53	3.55
Limestone	80.0	Total phosphorus	0.62	0.63
Salt	2.0	Non-phytate phosphorus	0.33	0.33
Premix ^1^	48.0	Lysine	0.82	0.82
Total	1000.0	Methionine + Cystine	0.69	0.69

^1^ Premix provided the following per kg of diet: Cu, 10 mg; Fe 50 mg; Mn 60 mg; Zn, 65 mg; Se 0.40 mg; retinyl acetate, 2.7 mg; cholecalciferol, 0.08 mg; tocopheryl acetate, 16.7; thiamin, 2.5 mg; riboflavin, 5 mg; cyano-cobalamin, 0.02 mg; biotin, 0.1 mg; folacin, 1 mg; pantothenic acid, 12 mg; nicotinic acid, 38 mg; and pyridoxine 3.5 mg.

**Table 2 animals-11-00701-t002:** The main physical parameters of the mash.

Index	Bulk Density, g/cm^3^	Apparent Density, g/cm^3^	Repose Angle, deg	Size Distribution, %
Value	0.71 ± 0.02	0.76 ± 0.01	36.01 ± 1.42	34.18 ± 1.06

**Table 3 animals-11-00701-t003:** The main physical parameters of the pellet.

Index	Value
Moisture content, %	5 ± 0.3
Post-pelleting temperature, °C	65 ± 1.4
Bulk Density, kg/m^3^	568 ± 4.3
Specific density, kg/m^3^	1455 ± 110.8
Hardness, N	87 ± 6.1
Water resistant time, s	6 ± 0.9
Diameter, mm	4 ± 0.2
Length, mm	8 ± 0.7
Fine Powder like content, %	3 ± 0.3

**Table 4 animals-11-00701-t004:** Effects of feed form on the production performance of layers from 26 to 43 weeks of age.

Item	Hy-Line Grey		Hy-Line Brown		*p*-Value		
	Mash, *n* = 54	Pellet, *n* = 54	Mash, *n* = 54	Pellet, *n* = 54	Feed Form (F)	Breed (B)	F × B
Laying rate, %	88.19 ± 1.22 ^c^	91.66 ± 2.32 ^b^	91.26 ± 2.83 ^b^	94.10 ± 1.85 ^a^	<0.01	<0.01	<0.01
FCR ^1^	2.14 ± 0.11 ^a^	2.03 ± 0.09 ^b^	1.91 ± 0.10 ^c^	2.03 ± 0.12 ^b^	<0.01	0.03	<0.01
ADFI ^2^, g	111.76 ± 5.67 ^b^	115.96 ± 6.64 ^a^	114.63 ± 6.21 ^a^	117.19 ± 5.82 ^a^	0.13	0.14	0.55
Abnormal egg, %	1.60 ± 0.49	0.49 ± 0.37	1.37 ± 0.79	0.68 ± 0.46	0.57	0.12	0.24
Mortality, %	1.85 ± 0.25	1.85 ± 0.25	0.00 ± 0.00	1.85 ± 0.25	0.98	0.76	0.37

^1^ Feed conversion ratio, ^2^ average daily feed intake; ^a–c^ Means with different superscripts within each row are significantly different (*p* < 0.05).

**Table 5 animals-11-00701-t005:** Effects of feed form on egg quality traits of layers from 26 to 43 weeks of age.

Item	Hy-Line Grey		Hy-Line Brown		*p*-Value		
	Mash, *n* = 240	Pellet, *n* = 240	Mash, *n* = 240	Pellet, *n* = 240	Feed Form (F)	Breed (B)	F × B
Egg weight, g	59.54 ± 1.30 ^b^	62.13 ± 1.68 ^a^	62.58 ± 1.60 ^a^	63.16 ± 1.24 ^a^	0.03	0.12	0.07
Egg shape index	1.28 ± 0.26	1.30 ± 0.20	1.29 ± 0.14	1.30 ± 0.13	0.15	0.18	0.46
Shell thickness, mm	0.43 ± 0.01	0.43 ± 0.01	0.43 ± 0.02	0.42 ± 0.02	0.86	0.57	0.37
Shell strength, kg/cm^2^	4.15 ± 0.71	4.25 ± 0.86	3.70 ± 0.70	3.78 ± 0.97	0.36	<0.01	<0.01
Egg shell, %	12.70 ± 1.74	12.53 ± 3.12	12.62 ± 1.53	12.79 ± 1.45	0.99	0.62	0.35
Egg yolk, %	27.90 ± 5.93	28.23 ± 4.87	24.93 ± 3.60	25.42 ± 5.52	0.19	<0.01	0.23
Yolk color	4.22 ± 0.45	4.10 ± 0.62	6.0 ± 1.13	6.1 ± 1.22	0.92	<0.01	<0.01
Haugh unit	86.83 ± 5.66	88.12 ± 6.71	85.74 ± 5.23	89.59 ± 4.30	0.49	0.74	0.38

^a,b^ Means with different superscripts within each row are significantly different (*p* < 0.05).

**Table 6 animals-11-00701-t006:** Effects of feed form on the apparent digestibility of nutrients of layers.

Item	Hy-Line Grey		Hy-Line Brown, *n* = 108		*p*-Value		
	Mash, *n* = 6	Pellet, *n* = 6	Mash, *n* = 6	Pellet, *n* = 6	Feed Form (F)	Breed (B)	F × B
Dry matter, %	82.15 ± 2.87 ^b^	84.39 ± 2.94 ^a^	81.58 ± 1.88 ^b^	83.36 ± 2.38 ^a^	0.04	0.31	0.33
Crude protein, %	57.05 ± 1.83 ^b^	60.08 ± 2.17 ^a^	56.73 ± 1.77 ^b^	59.33 ± 2.01 ^a^	0.01	0.75	0.03
Total phosphorus, %	46.92 ± 2.03	47.46 ± 1.07	46.79 ± 1.47	45.77 ± 1.11	0.64	0.63	0.26
Calcium, %	56.71 ± 1.37	57.19 ± 1.44	56.34 ± 0.88	55.16 ± 1.04	0.15	0.13	0.56

^a,b^ Means with different superscripts within each row are significantly different (*p* < 0.05).

**Table 7 animals-11-00701-t007:** Effects of feed form on the intestinal morphology structure of layers.

Intestinal Parts	Item ^1^	Hy-Line Grey		Hy-Line Brown		*p*-Value		
		Mash, *n* = 6	Pellet, *n* = 6	Mash, *n* = 6	Pellet, *n* = 6	Feed Form (F)	Breed (B)	F × B
Duodenum	VH, μm	1432.52 ± 212.37 ^b^	1871.31 ± 230.15 ^a^	1393.91 ± 233.14 ^b^	1786.24 ± 215.60 ^a^	<0.01	0.39	<0.01
	CD, μm	131.39 ± 12.16 ^b^	145.83 ± 14.20 ^a^	140.92 ± 16.80 ^ab^	141.33 ± 13.42 ^a^	<0.01	0.26	0.25
	VCR	11.43 ± 1.16 ^b^	13.14 ± 1.05 ^a^	10.14 ± 1.46 ^b^	13.91 ± 1.77 ^a^	<0.01	0.18	0.23
Jejunum	VH, μm	1928.82 ± 228.32	2018.04 ± 265.43	1831.33 ± 331.03	1839.9 ± 236.66	0.74	0.27	0.67
	CD, μm	169.87 ± 14.91	170.52 ± 15.88	139.79 ± 18.57	157.04 ± 22.6	0.48	0.09	0.51
	VCR	11.81 ± 1.09	12.44 ± 1.27	13.92 ± 2.80	12.43 ± 2.11	0.68	0.36	0.37
Ileum	VH, μm	1356.04 ± 208.64 ^b^	1823.52 ± 227.85 ^a^	1622.22 ± 198.24 ^a^	1690.12 ± 222.20 ^a^	<0.01	0.03	<0.01
	CD, μm	124.03 ± 9.87	130.36 ± 10.96	113.44 ± 14.50	141.08 ± 12.41	0.57	0.61	0.24
	VCR	11.39 ± 1.21 ^b^	15.27 ± 1.37 ^a^	14.53 ± 2.82 ^a^	12.11 ± 2.21 ^ab^	0.03	0.71	0.28

^1^ VH, villus height; CD, crypt depth; VCR, ratio of villus height to crypt depth; ^a,b^ Means with different superscripts within each row are significantly different (*p* < 0.05).

## Data Availability

Not applicable.

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
