# Peer review of "Effect of the Pellet and Mash Feed Forms on the Productive Performance, Egg Quality, Nutrient Metabolism, and Intestinal Morphology of Two Laying Hen Breeds"

_animals, 2021, doi:10.3390/ani11030701_

Round 1
Reviewer 1 Report
Comments to the Authors of manuscript number: animals-1103647 entitled “Effect of the Pellet and Mash Feed Forms on the Productive Performance, Egg Quality, Nutrient Metabolism, and Intestinal 3 Morphology of Two Laying Hen Breeds”.
Authors presented the study performed on laying hens which were fed two different forms of feed, in pellet or mash form. Next, the difference in laying rate, feed intake, egg albumen quality and apparent metabolic rate of laying hens were determined. The study is very well planned and described. The manuscript is correctly written as a logical whole. It is a pleasure to read. There is some missed information in the methods description, however the study and paper are very good.
I recommend this manuscript to publication but after small correction. It also fits to chosen Journal-Animals.
- How were fed layers during preliminary period? It should be explained.
- L 105 – chickens or hens?
- How many hens were in one cage? It should be given
- If in one cage were only 2 hens, how laying rate was calculated? It should be explained
- How were calculated other data if in one cage were 2 or more hens?
- L 139 – in order: duodenum, jejunum and ileum
- L 141 – common histological procedure is quite different, the way from formalin to paraffin it is to simple presented
- L 142 - How villus was determined?
- L 143 - How crypth depth was determined and measured?
- Is possible to give the body weight? Maybe it is worth to present the body weight from this study.
- L 174- chicken?
- When egg quality was determined? Within how many hours from the lay by the hens? Was the egg fresh?
- Table 5, L 114 – was this shell strength a crushing force? How this determination look like? Was each egg at the same position? What axis of the egg acted the force in?
- L 213 Authors very well provided their discussion. They also gave probably reason – facilitated feed intake.
- L 242 – CD does not participate in the calculation of surface are of intestine. The ratio VH/CD indicated secretory ability.
Reviewer 2 Report
Reviewer Comments:
The study analyzed whether Hy-Line brown and Hy-Line gray hens differ in the performance and in the blood biochemical characteristics when feed pelleted fed. It is an interesting study. However, several major and minor comments are necessary, because lack several clarifications and/or additional information in the material and methods, as well as there is problem with the statistical analysis that compromise the manuscript quality.
Comments:
First at all, please use “hens” or “layers” instead of “chickens”.
Statistics: What was the form of the applied GML model? Why (line)x(feed_form) interaction was not included? You disuse this aspect (different response for each line) intensively). Were t-test assumption checked (data and variance distribution)? Isn’t 6 birds (one per replication cage) per treatment a little bit too little for histological examination? Was analysis of test power performed? Are the results of egg quality examination pooled through the whole experimental period? Also the information on n number for each measurement is needed (both for performance, general characteristic of eggs, intestine morphology as well as pellets physical traits).
How physical parameters of pellets were established? Short information (equipment, methods) in supplementary materials should be added, as these are crucial information.
Minor comments:
L83 “... at 20.3C and 56%, respectively...”
Table 2 Please change unit of repose angle to “deg” (degree). Also that is “unit weight” ? It has a unit of density (g/cm3).
L103 Hens form experimental groups were fed pellets during pre-experimental period, am I right ?
L117 what yolk scale was used?
L131-133 Methods for DM, CP, Ca, and P determinations?
L142 Photos were taken using soft(ware)?
L154-156 Remove
Table 5 – Please use SI unit for shell strength (N). Kilogram per square centimetre (kg/cm2) is a unit of pressure (old unit of technical atmosphere). How this value was calculated?
L208-209 Higher feed intake of pelleted fed was found only for Hy-line grey breed.
L240 Is the small intestine the main organ of the GIT? On what basis? Please rephrase.
References: Please correct double numbering.
Reviewer 3 Report
Overall while this research is interesting, it is very difficult to evaluate this research due to a lack of important information.
Major points:
- Given the short duration of this experiment, and relatively few number of birds per treatment, I believe it is necessary to show weekly feed intake and egg production data so the reader can assess how quickly these changes occurred. Currently you do not provide data indicating that within the Hy-Line grey and Hy-Line brown groups that the hens fed the mash or pelleted diets started with equal body weights, feed consumption, and egg production at the start of the experiment. Given the lower egg production when feeding the mash diets, having body weights would be nice as well.
- Line 81 indicates that the hens were fed 2 times each day. Did they have free access to feed at all times or was the feed completely consumed between feedings? This is especially important to know with the mash diets, because if all the food was being eaten at least once a day the separation/selection of components of the mash diet as discussed in lines 226-227 would not be applicable.
- Lines 89-91 seem to indicate that once the mash diet was made, to make the pelleted diet water was added to the mash diet at a rate of 5% by weight. This would increase the moisture content of the pelleted diet. Thus if it contained more moisture diluting the nutrients, I would expect increased consumption of this diet to deliver the same amount of nutrients as the mash diet. So is your feed consumption data adjusted for the increased water weight of the pelleted diet?
- Line 91 and Table 1. I would expect the feed ingredients to be exactly the same not “similar” and the calculated analyses to be exactly the same. Each time the diets were made, they should have been mixed as one basal diet that was simply split with half remaining in mash form and the other half being pelleted.
- Tables 4-7, please indicate if the values are means plus or minus SD. In addition, please indicate the n value for each table. Based on the line 103, n would equal 6 for all of your analyses. If this is the case, then your SEM values would be all based on the square root of 6 which makes some of the statistical differences being significant such as ADFI questionable.
- Lines 229-238, for this part of the discussion you need to make more of a distinction between your results and references 24 -27. You did not provide any evidence that the corn particle size was different between your mash and pelleted diets. Your pellets are going to disintegrate in the digestive tract and feed component particle sizes will be the same between mash and pelleted diets. This cited previous research was based on using either whole grains or coarsely ground particles versus more finely ground particles when the pellets were made. So once the pellets disintegrate the large particles of the grains or whole grains were still present and caused the results cited.
- Although you indicate the hens had a 1-week acclimation period, I think it is necessary to indicate if the feed from was pellet or mash prior to this acclimation period.
Minor points:
- The mortality numbers do not make sense to me in Table 4. If you have 54 birds in each treatment and 1 bird dies that would equal 1.85%, so how did you have 0.01% for 3 of the 4 treatments?
- Line 31 replace the word biweekly as it can be interpreted as every 2 weeks or twice a week.
- Consider replacing apparent metabolic rate with apparent digestibility throughout the manuscript.
4. Lines 122-123, as written there is an implication that there was a 72-hour period to clear the digestive tract for the apparent digestibility studies. Typically, this period is 24 hours.
Round 2
Reviewer 1 Report
I have no more comments.
Author Response
Thank you for your suggestion.
Reviewer 2 Report
Thanks to the Authors for the improvements they made. However there are still some issues which I think should be clarified.
- Owing the not identical results observed in both lines (duodenum CD, ileum VH, ) and the discussion, where results are often discussed in line-specific manner, I still think that the applied model should include (line)x(feed type) interaction, and the differences should analyses using ANOVA with post-hoc test (Tukey's most preferably) rather than performing two independent t-test comparison for two breeds separately. Two-way ANOVA will not weaken the work in any way, on the contrary, it will allow to show which changes are line-specific and which are universal.
- Answer: “The physical parameters of mash and pellet were measured as previous described (Van Der Poel and Thomas, 1996).”
10.1016/0377-8401(96)00949-2 it is a review article. It does not describe any experimental protocols, some of the measured physical parameters of the pellet are not even mentioned (like water resistant time) in this review. Please name the used apparatus and equipment, describe in brief to experimental protocols (of give the appropriate reference).
- What was the form of the feed given to hens BEFORE the pre-experimental period?
- I don't think that eggs were measured within 2 h of being laid, I suppose that egg collection and measurements were performed once per day.
- What was the selection criteria of birds for digestibility studies? What was the amount of TiO2 given to the feed?
- Answer: “In order to determine analytical reliability and reduce experimental errors, experiment was suggested to be repeated with at least 3 to 6 biological preparations. Therefore, six replicates for each treatment were supposed to be sufficient for histological examination in the present study”
Who suggested that the experiment should be repeated with at least 3 to 6 biological preparations? What do you mean be “biological preparations”? In histological examinations results obtained for single experimental unit (hen) that are subjected to statistical analysis generally are averages of multiple measurements or analysed as repeated measurements. How many measurements of VH and CD per experimental hens were performed?
- Answer: “Yes, it has been written as software (Line 155).”
Motic BA210 is a microscope, not a software. Stained slides were observed under a Motic BA210 and the images were taken/analysed using Motic software.
L101 I don’t think that term “hens have been homogenized” is a proper one.
Reviewer 3 Report
Thank you for revising the manuscript. Some of my original concerns remain.
- I still believe given the short duration of this experiment, and relatively few number of birds per treatment, that it is necessary to show feed intake and egg production data weekly or even every 2 weeks. Graphs of egg production or even feed intake would not be too complex to show. At 26 weeks feed intake and egg production are equal, but by 43 weeks they were likely not based on your overall data. It is important to know if this difference occurred gradually or was more pronounced later in the experiment due to say intestinal morphological changes.
- The lack of the body weight data is problematic. Even if you do not have body weight data for the whole experiment do you have the end body weights at the conclusion of the experiment?
- While your response to me indicates not all of the feed was consumed between feedings please indicate this in the manuscript as this could indicate feed separation and selection was occurring.
- In your response to me, you indicate that to make the pelleted diet water was added to the mash diet at a rate of 5% by weight and that this increased the moisture content. You then indicate that the feed consumption data was adjusted to correct for this, but this is not detailed in the revised manuscript. The reason I originally brought this point up is that your increase in feed consumption for the pelleted diets is about 3 to 4% which could reflect the 5% dilution of nutrients by water.
- For Table 1 replace nutrient level with analyzed nutrient level. Indicate in the manuscript that when the diets were made, they were mixed as one basal diet, that was then simply split with one half remaining in mash form and the other half being pelleted.
- While I appreciate your change in the discussion of the revised manuscript, I am not sure that lines 242-247 are appropriate, as you are still trying to justify your results based on previous particle size results and that is not equivalent to your research.
- Although you indicate the hens had a 1-week acclimation period, I think it is necessary to indicate if the feed from was pellet or mash prior to this acclimation period. Knowing if the hens were being fed pellets or mash at the commercial farm prior to this experiment is important as one group had an adjustment to feed form and the other one did not.
8. The revised manuscript indicates that titanium dioxide (line 139) was used. This is not indicated in the diets and is not consistent with clearing out the digestive tract, which you added in response to my original comment or with the total feces collection that you indicate occurred in line 127
